# Serum Essential Elements and Survival after Cancer Diagnosis

**DOI:** 10.3390/nu15112611

**Published:** 2023-06-02

**Authors:** Jan Lubiński, Marcin R. Lener, Wojciech Marciniak, Sandra Pietrzak, Róża Derkacz, Cezary Cybulski, Jacek Gronwald, Tadeusz Dębniak, Anna Jakubowska, Tomasz Huzarski, Milena Matuszczak, Katherine Pullella, Ping Sun, Steven A. Narod

**Affiliations:** 1Department of Genetics and Pathology, International Hereditary Cancer Center, Pomeranian Medical University in Szczecin, ul. Unii Lubelskiej 1, 71-252 Szczecin, Poland; marcin.lener@pum.edu.pl (M.R.L.); sandra.pietrzak@pum.edu.pl (S.P.); cezary.cybulski@pum.edu.pl (C.C.); jgron@pum.edu.pl (J.G.); debniak@pum.edu.pl (T.D.); aniaj@pum.edu.pl (A.J.); huzarski@pum.edu.pl (T.H.); matuszczakmilena@gmail.com (M.M.); 2Read-Gene, Grzepnica, ul. Alabastrowa 8, 72-003 Dobra (Szczecińska), Poland; wojciech.marciniak@read-gene.com (W.M.); roza.derkacz@gmail.com (R.D.); 3Department of Clinical Genetics and Pathology, University of Zielona Góra, ul. Zyty 28, 65-046 Zielona Góra, Poland; 4Department of Nutritional Sciences, University of Toronto, Toronto, ON M5S 1A8, Canada; katherine.pullella@mail.utoronto.ca; 5Women’s College Research Institute, Toronto, ON M5S 1B2, Canada; ping.sun@wchospital.ca (P.S.); steven.narod@wchospital.ca (S.A.N.); 6Dalla Lana School of Public Health, University of Toronto, Toronto, ON M5T 3M7, Canada

**Keywords:** copper, zinc, selenium, breast cancer, laryngeal cancer, lung cancer, prostate cancer, survival

## Abstract

In a prospective study, we measured the associations between three serum elements (Se, Zn and Cu) and the prognosis of 1475 patients with four different types of cancer (breast, prostate, lung and larynx) from University Hospitals in Szczecin, Poland. The elements were measured in serum taken after diagnosis and prior to treatment. Patients were followed from the date of diagnosis until death from any cause or until the last follow-up date (mean years of follow-up: 6.0–9.8 years, according to site). Kaplan–Meier curves were constructed for all cancers combined and for each cancer separately. Age-adjusted hazard ratios (HRs) were estimated using Cox regression. The outcome was all-cause mortality. A Se level in the highest quartile was also associated with a reduced mortality (HR = 0.66; 95%CI 0.49–0.88; *p* = 0.005) in all-cause mortality for all cancers combined. Zn level in the highest quartile was also associated with reduced mortality (HR = 0.55; 95%CI 0.41–0.75; *p* = 0.0001). In contrast, a Cu level in the highest quartile was associated with an increase in mortality (HR = 1.91; 95%CI 1.56–2.08; *p* = 0.0001). Three serum elements—selenium, zinc and copper—are associated with the prognosis of different types of cancer.

## 1. Introduction

There is increasing interest in the role of serum/blood elements, including metals and metalloids, in the survival of cancer patients [1]. Cancers are heterogenous, but many cancers share genetic and epigenetic features in common and cancers at different sites and with different histologies may respond to the same chemotherapy. The fate of a patient with cancer is determined, in part, by the genetic and epigenetic makeup of the tumor itself (cell autonomous), but there are host factors as well (local and macro-environment) that influence the outcome (non-cell autonomous) [2]. Notable examples include tumor-infiltrating lymphocytes [3], the microbiome [4] and platelets [5]. It is possible that the metastatic niche permits the metastases and growth of more than one tumor type, and the metastatic potential of different cancers is under the influence of common factors. Among host factors, blood levels of several micro-elements have been implicated in cancer incidence and progression. We have previously reported on the effects of selenium, zinc and copper on the rates of cancer incidence [6,7,8] and cancer progression [9,10,11,12] in patients from our institution, but these studies have been site-specific.

### 1.1. Selenium

In a human body, selenium (Se) combines with proteins that are incorporated in the form of selenocysteine. As a component of selenoproteins, selenium plays both enzymatic and structural roles [13]. Some of the important functions of selenoproteins are the production of thyroid hormones, the stimulation of the immune system and protection against oxidative stress [14]. Reports on humans have matched well with the mechanisms of selenium activities described in animals [15,16]. Both a deficiency and excess of selenium can have adverse effects on the body, including, e.g., cardiac disorders, hypertension, reduced efficiency of the immune system, thyroid function disorders, disorders of bone mineralization, malformation of teeth and increased risks of cancer [17,18,19,20,21]. There are few studies which have evaluated the effect of selenium levels on cancer survival. Individual reports indicate that better survival is associated with higher serum selenium levels in patients with cancers of the breast, colon, malignant melanoma, larynx, lung and renal cell carcinoma [9,22,23,24,25,26,27,28].

### 1.2. Zinc

Zinc (Zn) is an essential element necessary for health which offers protection against free radicals. Zinc is a component of superoxide dismutase (SOD-2). It is also involved in the immunological processes and the function of the skin and mucous membranes. Zinc supports the storage and secretion of insulin from the pancreas and maintains normal levels of other micronutrients, including selenium, magnesium and copper. Zinc plays a role in the detoxification of heavy metals [29]. It has been observed that zinc accumulates in some cancer cells [30]. Normal prostate epithelial cells accumulate zinc; however, in prostate cancer cells, zinc concentration is reduced [31]. It has been proposed that zinc has anticancer effects, inhibiting the growth of cancer cells and activating apoptosis. Some studies have reported serum zinc levels to be higher in cancer patients than in others [32,33,34], whereas other studies have reported that it is lower [35]. Some studies suggest that dietary zinc has chemo-preventive properties. People with a diet rich in zinc have a lower risk of lung and colorectal cancers [36,37]. Moreover, some studies suggest that high dietary zinc intake [38] and higher zinc levels in the serum correlate with better survival in patients with prostate [38] and laryngeal cancers [26].

### 1.3. Copper

Copper (Cu) is an essential element that is involved in many physiological processes [5,6,7,8,9] and is a cofactor for numerous biological processes. Copper is a component of several metalloenzymes, including metalloproteinases, and plays roles in angiogenesis and oxidative phosphorylation. Several studies have reported high serum/blood copper levels in patients with prostate cancer [10,11,12] and other cancers, including lymphoma, reticulum cell sarcoma, laryngeal carcinomas, cervical, breast, pancreas, stomach and lung cancers [39,40,41]. We have recently shown that a high copper level was associated with an increased risk for the development of colon cancer [42]. In patients with breast and colorectal cancer, serum copper levels were correlated with disease stage [42,43]. There are few studies on the impact of circulating copper and cancer survival [44,45].

In this study, we evaluated the serum levels at diagnosis of three circulating elements as prognostic markers for four common cancers. We hypothesized that the elements’ levels are associated with survival are similar irrespective of the cancer site.

## 2. Materials and Methods

### 2.1. Study Group

We studied 1475 patients with four different types of cancer (breast, prostate, lung and larynx). All cases were unselected patients treated between 2009 and 2018 at the Clinical Hospitals of Pomeranian Medical University in Szczecin, Poland, or at an associated hospital. All patients were seen at these hospitals for the period specified and were enrolled into the study. Typically, these patients are offered genetic testing shortly after diagnosis during an out-patient visit to one of our clinics and are offered the opportunity to participate in other clinical research studies. Blood samples were taken from cases between 2009 and 2018 shortly after diagnosis and prior to treatment. Patients were restricted to those diagnosed at age 85 and younger. Medical charts were reviewed for date of diagnosis, smoking status (yes/no) and chemotherapy (yes/no). Additional information was collected for breast cancer (ER status, nodal status and tumor size) and for prostate cancer (Gleason grade, PSA (prostate-specific antigen) at diagnosis and prostatectomy (yes/no)). Tumor stage was assigned to one to four stages for larynx and lung cancer. The study was conducted in accordance with the Helsinki Declaration and with the consent of the Ethics Committee of Pomeranian Medical University in Szczecin under the number KB-0012/73/10 on 21 June 2010. All participants provided written informed consent to be enrolled in the study.

### 2.2. Sample Collection, and Storage and Measurement of Elements

Patients were asked to fast for six hours before sample collection. A total of 10 mL of peripheral blood was collected into a Beckton Dickinson Vacutainer tube [REF.367953] containing a clot activator. After the collection, the tubes were incubated at room temperature for a minimum 30 min to clot, but no longer than 120 min, and after which the tubes were centrifuged at 1300× *g* for 12 min. After the centrifugation, the serum was transferred into cryovials and deposited at −80 °C until analysis. On the day of analysis, the sera were thawed, vortexed and centrifuged at 5000× *g* for 5 min.

Determination of selenium (^78^Se), zinc (^66^Zn) and copper (^65^Cu) was performed using an ICP mass spectrometer ELAN DRC-e (PerkinElmer). Before each analytical run, the instrument was tuned to meet the manufacturer’s criteria. Oxygen was used as a reaction gas. Technical details are available upon request. The spectrometer was calibrated using external calibration. Calibration standards were prepared fresh daily from 10 µg/mL of Multi-Element Calibration Standard 3 (PerkinElmer, Waltham, MA, USA) by diluting it with blank reagent to the final concentration of 1, 2, 5, 10, 50 and 100 µg/L for Se, Zn and Cu determination. Correlation coefficients for calibration curves were always greater than 0.99. Matrix-matched calibration was used. The analysis protocol assumed 30-fold dilution of serum in a blank reagent. The blank reagent consisted of high-purity water (>18 MΩ), TMAH (Alfa Aesar, Tewksbury, MA, USA), Triton X-100 (PerkinElmer), n-butanol (Merck, Darmstadt, Germany) and EDTA (Sigma-Aldrich, St. Louis, MO, USA) [46,47,48]. The accuracy and precision of measurements were tested using the certified reference material (CRM) Clinchek Plasmonorm Serum Trace Elements Level 1 (Recipe, Munchen, Germany).

### 2.3. Statistical Analysis

For each of the three elements, patients were assigned to one of three categories based on the distribution of the values for the elements in the entire dataset. The cutoffs were low (<25th percentile), middle (25th to 75th percentile) and high (75th percentile). Cutoff values were determined separately for men and for women.

Patients in each of the four cancer site groups were compared based on a range of variables, including age, sex, the median level of each of the elements and five-year survival. We also conducted a correlation analysis for the levels of elements (two by two) using Pearson’s correlation coefficient.

The principal outcome was death from any cause. Death was ascertained by linking to the Polish Statistics Registry. Patients were followed for all-cause survival at diagnosis using survival analyses. Five-year survival was estimated for each site using the Kaplan–Meier method and crude statistical differences were assessed using the log-rank test. Statistical significance was set at *p* < 0.05 (two-sided).

To estimate the association of each of the three elements with survival in all patients combined, hazard ratios (HRs) and 95% confidence intervals (CIs) were calculated using the Cox proportional hazards model. Patients were followed from the date of diagnosis to death or 1 January 2022. For each comparison the middle category was defined as the reference group. The initial analysis included all patients. The basic model was adjusted for age of diagnosis only. In the multivariable model, outcomes were adjusted for age at diagnosis, cancer site, current smoking status (yes/no), sex and the other two elements. The HRs were estimated for low and high levels of selenium, zinc and copper, and were compared to the middle level. 

A series of site-specific analyses was then conducted. For each site, three analyses were conducted, one for each element. For prostate cancer, the multivariable HRs were adjusted for age at diagnosis, PSA (four levels) and Gleason grade (<7; 7+). For breast cancer, the multivariable HRs were adjusted for age at diagnosis, ER status, tumor size and nodal status (positive/negative). For lung and laryngeal cancer, the HRs were adjusted for sex, stage, age at diagnosis and current smoking status. Statistical analyses were conducted in SAS version 9.4.

## 3. Results

A total of 1475 cancer patients were enrolled in this study (794 men and 681 women). The patient characteristics are presented in Table 1. The mean levels of selenium, zinc and copper were compared with regard to cancer type (Table 1), age and sex (Table 2).

Across all age categories, zinc and selenium levels were higher in women than in men (Table 2). Copper levels were similar in men and women. There was little variation in the elements by age (Table 2). The cutoff levels for percentiles for men and women corresponding to the low, middle and high categories are presented in Table 3.

The mean levels of zinc were lower in patients with laryngeal and lung cancer than in patients with other cancer types. Only 7% of lung cancer patients and 3% of laryngeal cancer patients had a zinc level in the high category. The correlations between the serum levels of the three elements are presented in Table 4. There was a modest positive correlation between the serum zinc and selenium levels.

Five-year survival rates were 45% for lung cancer, 63% for laryngeal cancer, 83% for prostate cancer and 86% for breast cancer. The crude five-year survival rates by cancer site and serum levels of the three elements are presented in Figure 1, Figure 2, Figure 3 and Figure 4.

In the basic (age-adjusted) model, we sought to see if serum elements were predictive of survival in all cancer sites combined. In general, high levels of all three elements were predictive of survival. The associations persisted after adjusting for cancer site, age at diagnosis, smoking and age (Table 5). A low zinc level was associated with relatively poor survival and a high zinc level was associated with a relatively good survival. A low selenium level was associated with poor survival and a high selenium level was associated with good survival. In contrast, a low copper level was associated with good survival and a high copper level was associated with poor survival.

HRs for the association between the three elements and all-cause mortality for the individual sites are presented in Table 6, Table 7, Table 8 and Table 9. 

For lung cancer, the associations were modest and were attenuated after adjusting for tumor stage (Table 6).

For laryngeal cancer, a low selenium level was predictive of poor survival in both the crude data (Figure 2C) and after adjusting for stage (Table 7). For each of the three elements, the quartile levels were based on the distribution of the entire patient cohort and were divided by sex. For both lung and laryngeal cancer there were few subjects with a zinc level or selenium level in the highest quartile, and for these evaluations the top three quartiles were merged. 

For breast cancer and prostate cancer, the associations with all three elements were similar (Table 8 and Table 9). For breast cancer, both a high copper and a low selenium level were predictive of poor survival (Table 8, Figure 3B,C). There was also a trend related to poor survival with lower levels of zinc (Figure 3A). Similar effects were seen with prostate cancer (Table 9, Figure 4). For breast and prostate cancer, the HRs were attenuated slightly after adjusting for grade and the other elements.

## 4. Discussion

In this study, we evaluated the serum levels at diagnosis of three circulating elements as prognostic markers for various common cancers. The current study updates our previous reports for selenium and zinc [7,8,9,22,23,25,27] and extends the follow-up time by linking to the Polish Vital Statistics Registry. We have also added new data for copper. 

In general, we can see that high selenium and zinc levels were predictive of a good prognosis and high copper levels were detrimental, but the associations and trends varied across the four cancer sites and were attenuated after adjusting for tumor stage. We report strong associations between a high copper level and poor survival for both breast and prostate cancer. The associations were present in the age-adjusted analysis and persisted after adjustment for other prognostic factors. A low serum selenium level was also associated with a poor prognosis for breast cancer patients, as reported previously [22,23].

Serum selenium and zinc predicted mortality for laryngeal and lung cancer patients, but the associations were modest and were attenuated after adjusting for stage [9,27]. Selenium and zinc levels in patients with larynx and lung cancer were low overall, suggesting that these elements play important roles in lung and laryngeal cancer incidence more than in cancer survival [26,27,49]. There were too few patients with laryngeal or lung cancer with high zinc or selenium levels to obtain precise HRs for these patient groups. 

There are several limitations to our study. We did not include patients representing all cancer sites; notably, we did not include patients with gastrointestinal or gynecologic cancers. These will be the subject of future studies. We have recently reported that elevated blood copper was associated with the presence of colon cancer [8], but it is too early to assess survival in our cohort. Nevertheless, our patient population represented three of the four most common sites in Poland. We did not have details on the cause of death because the study was conducted by vital status linkage and the cause of death is not available in the Polish Death Registry. There are many factors which influence prognosis and there may be unmeasured confounding factors as well. Possible confounders include family histories, dietary supplements, comorbidity, exercise and BMI. Ongoing studies are underway to assess the ten-year follow up, to expand the range of cancer sites as well as to determine the specific cause of death.

The strength of this study is the large sample of patients (*n* = 1475), and all three elements were measured in all patients using standard methods in a single laboratory. For consistency and simplicity, we divided patients into three categories representing low, middle and high. These are constructed distinctions and are not intended to reflect current standards or recommendations. We elected to use the highest 75th percentile as the high category. This was not based on pre-existing standards but was chosen for statistical power and expediency. However, by using this definition very few of the lung and laryngeal cancer patients fell into the high category. 

We measured the elements in the serum after diagnosis but prior to treatment. This method was chosen to exclude the effect of treatments on serum element levels. However, it would be of interest to see if changes in the serum level of copper and the other elements post-treatment are reflective of the presence of residual disease, metastatic disease, and/or if they predict prognosis post-treatment. If so, they may serve as useful tumor markers, as is the case for CA125 [50,51], PSA [52], platelets [53] and circulating tumor DNA [54].

The associations here are not proof of causality, but are hypothesis generating. The specific mechanisms by which the metals influence prognosis are uncertain. The levels of different metals may be a marker of risk, but not causally in the progression pathway. Selenium, zinc and copper are all involved in many metabolic processes that can impact prognosis. It is possible that the driver of progression is a conjugate protein activity and the metal level may be a consequence of the underlying physiological process—such that homeostasis is altered—with a resultant change in the serum level of the element. To provide a brief overview, selenium compounds can generate oxygen free radicals and various selenoproteins, including glutathione peroxidase 1–4 and 6, as well as thioredoxin reductase, which have also been linked to neoplastic growth [55,56,57]. Zinc is known to be involved in oxidative stress (including zinc/copper superoxide dismutase), DNA repair (replication and transcription via zinc finger proteins and controlling DNA binding activity, such as AP-1, NFκB and p53), cell signaling and apoptosis (also via NFκB and AP-1) [58,59]. Finally, copper plays a critical role in cellular oxygenation, the neutralization of free radicals, angiogenesis and cellular iron metabolism. A novel pathway of copper-dependent cell-death, cuproptosis [60,61], may also be linked to cancer prognosis. 

The sheer number of proteins and processes which involve these metals present a challenge to clarify the mechanism. Nevertheless, we are encouraged to pursue clinical studies directed at lowering the serum copper level and raising the levels of zinc and selenium in selected patients. Zinc and selenium can be given as supplements, but to lower copper, it is necessary to give a chelating agent such as penicillamine or TNT. Ultimately, the utility of the treatment will depend on the mechanism of action. If the elements facilitate metastatic spread, it may be challenging because for breast and other cancers, the first progression (metastases) may precede the diagnosis. However, if the elements promote the growth of cancer cells within the metastatic niche or prevent grade progression, then providing the therapy post-diagnosis may be appropriate. 

## 5. Conclusions

Three serum elements—selenium, zinc and copper—are associated with the prognosis of different types of cancer. A high copper level can be associated with an increase in all-cause mortality for the studied cancers (breast, prostate, lung and larynx). In contrast, high zinc and selenium levels can be associated with reduced mortality. Clinical studies directed at lowering serum copper levels and raising the levels of zinc and selenium in selected cancer patients are required.

## Figures and Tables

**Figure 1 nutrients-15-02611-f001:**
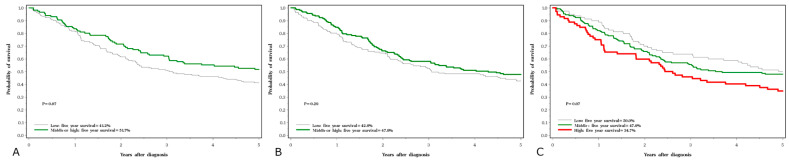
Five-year survival associated with three element levels in lung cancer subjects; (**A**) selenium, (**B**) zinc, (**C**) copper.

**Figure 2 nutrients-15-02611-f002:**
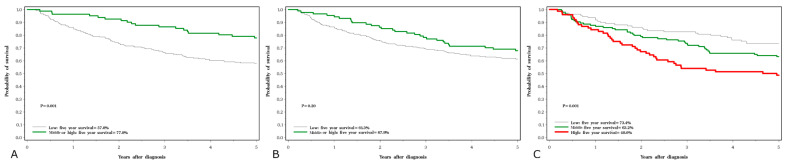
Five-year survival associated with three element levels in larynx cancer subjects; (**A**) selenium, (**B**) zinc, (**C**) copper.

**Figure 3 nutrients-15-02611-f003:**
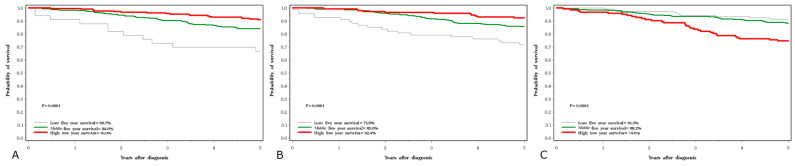
Five-year survival associated with three element levels in breast cancer subjects; (**A**) selenium, (**B**) zinc, (**C**) copper.

**Figure 4 nutrients-15-02611-f004:**
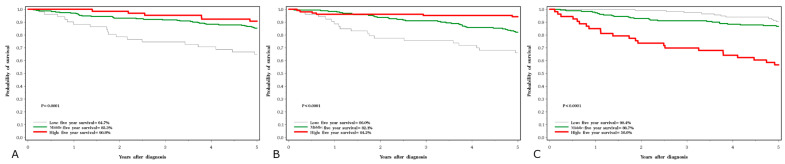
Five-year survival associated with three element levels in prostate cancer subjects; (**A**) selenium, (**B**) zinc, (**C**) copper.

**Table 1 nutrients-15-02611-t001:** Characteristics of the 1475 patients in the study.

	Cancer Site			
	Breast *n* = 531	Prostate *n* = 347	Lung *n* = 298	Larynx *n* = 299
Age of diagnosis				
Years (mean, range)	57.2 (26–84)	66.0 (41–84)	64.1 (43–84)	61.0 (40–81)
Sex				
Male	0	347 (100%)	193 (65%)	254 (85%)
Female	531 (100%)	0	105 (35%)	45 (15%)
Year of diagnosis (mean, range)	2011 (2009–2016)	2013 (2010–2015)	2011 (2010–2012)	2013 (2010–2018)
Selenium level (µg/L) (mean, range)	86 (52–172)	78 (42–138)	62 (17–108)	58 (21–105)
Zinc level (µg/L) (mean, range)	867 (525–11045)	847 (516–1340)	718 (350–1071)	640 (358–1318)
Copper level (µg/L) (mean, range)	1153 (685–2153)	1093 (460–2197)	1146 (671–2866)	1116 (436–2795)
Years of follow-up (mean)	9.4	7.6	6.0	6.8
5- year survival	85.9%	83.3%	45.3%	63.2%

**Table 2 nutrients-15-02611-t002:** Mean levels of copper, zinc and selenium according to age and sex.

	Men (*n* = 794)			Women (*n* = 681)		
Age of Diagnosis Frequency M/F	Selenium (µg/L) Mean, Range	Zinc (µg/L) Mean, Range	Copper (µg/L) Mean, Range	Selenium (µg/L) Mean, Range	Zinc (µg/L) Mean, Range	Copper (µg/L) Mean, Range
<50 20/139	1213 (839–1545)	665 (377–884)	61 (31–92)	1150 (500–2049)	840 (525–1254)	83 (40–124)
50–60 221/227	1124 (643–1391)	729 (358–1244)	65 (21–138)	1158 (632–2151)	867 (455–11045)	83 (35–172)
60–70 376/213	1090 (436–2197)	761 (359–1340)	69 (25–109)	1172 (658–2866)	810 (350–1389)	80 (33–123)
70+ 177/102	1092 (460–1842)	745 (366–1190)	67 (17–106)	1161 (710–1910)	794 (553–1128)	76 (36–121)

**Table 3 nutrients-15-02611-t003:** Cutoff levels used to determine low, middle and high categories for each element by sex.

	Men (*n* = 794)			Women (*n* = 681)		
Cutoff Level	Selenium (µg/L)	Zinc (µg/L)	Copper (µg/L)	Selenium (µg/L)	Zinc (µg/L)	Copper (µg/L)
25 percentile	56	636	944	71	727	1011
75 percentile	78	850	1230	93	899	1260

**Table 4 nutrients-15-02611-t004:** Correlation coefficients for three elements studied (*n* = 1475).

	Selenium	Zinc	Copper
Selenium	-		
Zinc	0.195 *p* < 0.0001	-	
Copper	0.059 *p* = 0.13	0.087 *p* = 0.02	-

**Table 5 nutrients-15-02611-t005:** HRs associated with low and high serum metals all-cause mortality (all sites combined).

	Total /Deaths	Age-Adjusted Only	Adjusted
		HR (95%CI) *p*-Value	HR (95%CI) *p*-Value
Selenium			
Low	484/250	2.18 (1.81–2.62) < 0.0001	1.61 (1.28–2.02) < 0.0001
Middle	702/206	1.0	1.0
High	289/58	0.66 (0.49–0.88) 0.005	0.71 (0.52–0.96) 0.03
Zinc			
Low	473/224	1.68 (1.40–2.01) < 0.0001	1.19 (0.97–1.47) 0.10
Middle	737/240	1.0	1.0
High	265/50	0.55 (0.41–0.75) 0.0001	0.72 (0.52–0.99) 0.04
Copper			
Low	406/115	0.86 (0.69–1.33) 0.19	0.81 (0.64–1.01) 0.07
Middle	746/239	1.0	1.0
High	323/160	1.91 (1.56–2.08) < 0.0001	1.72 (1.41–2.11) < 0.0001
Current Smoker			
No	641/183	1	1
Yes	676/278	1.77 (1.46–2.13) < 0.0001	0.97 (0.76–1.25) 0.83
Missing	158/53	1.11 (0.81–1.51) 0.51	1.46 (0.99–2.15) 0.06
Sex			
Female	681/214	1	1
Male	794/300	1.28 (1.06–1.53) 0.009	1.21 (0.91–1.60) 0.18

Adjusted model adjusted for age, sex and current smoking status in addition to the other elements and cancer sites.

**Table 6 nutrients-15-02611-t006:** HRs associated with low and high serum metals and all-cause mortality (lung cancer subjects).

	Total /Deaths	Age-Adjusted Only	Adjusted
		HR (95%CI) *p*-Value	HR (95%CI) *p*-Value
Selenium			
Low	182/109	1.34 (0.97–1.85) 0.08	1.16 (0.81–1.66) 0.43
Middle	111/55	1.0	1.0
High	5/2	0.68 (0.17–2.80) 0.59	1.14 (0.27–4.85) 0.86

Zinc			
Low	141/83	1.25 (0.91–1.71) 0.17	1.17 (0.83–1.65) 0.37
Middle	145/74	1.0	1.0
High	12/9	1.73 (0.86–3.45) 0.12	1.33 (0.62–2.87) 0.47

Copper			
Low	80/41	0.88 (0.61–1.29) 0.52	0.97 (0.65–1.46) 0.88
Middle	146/78	1.0	1.0
High	72/47	1.39 (0.97–2.01) 0.07	1.19 (0.82–1.73) 0.36

Adjusted model adjusted for age, sex, stage and current smoking status in addition to the other elements.

**Table 7 nutrients-15-02611-t007:** HRs associated with low and high serum metals and all-cause mortality (larynx cancer subjects).

	Total /Deaths	Age-Adjusted Only	Adjusted
		HR (95%CI) *p*-Value	HR (95%CI) *p*-Value
Selenium			
Low	218/96	2.61 (1.51–4.50) 0.0006	2.19 (1.23–3.91) 0.008
Middle	72/15	1.0	1.0
High	9/5	3.32 (1.20–9.18) 0.002	2.10 (0.59–7.50) 0.26

Zinc			
Low	212/87	1.46 (0.94–2.26) 0.09	1.03 (0.65–1.65) 0.89
Middle	82/26	1.0	1.0
High	5/3	2.35 (0.71–7.79) 0.16	1.46 (0.31–6.86) 0.63

Copper			
Low	109/32	0.71 (0.45–1.12) 0.14	0.95 (0.59–1.53) 0.83
Middle	114/44	1.0	1.0
High	76/40	1.64 (1.06–2.52) 0.003	1.15 (0.74–1.79) 0.54

Adjusted model adjusted for age, sex stage and current smoking status in addition to the other elements.

**Table 8 nutrients-15-02611-t008:** HRs associated with low and high serum metals and all-cause mortality (breast cancer subjects).

	Total /Deaths	Age-Adjusted Only	Adjusted
		HR (95%CI) *p*-Value	HR (95%CI) *p*-Value
Selenium			
Low	33/21	2.43 (1.49–3.97) 0.0004	2.55 (1.52–4.28) 0.0004
Middle	288/82	1.0	1.0
High	210/40	0.64 (0.44–0.94) 0.02	0.69 (0.46–1.03) 0.07

Zinc			
Low	67/29	1.52 (0.99–2.34) 0.05	1.37 (0.87–2.18) 0.18
Middle	320/89	1.0	1.0
High	144/25	0.58 (0.37–0.90) 0.02	0.72 (0.46–1.14) 0.16

Copper			
Low	103/21	0.83 (0.51–1.34) 0.44	0.87 (0.52–1.44) 0.59
Middle	306/78	1.0	1.0
High	122/44	1.70 (1.17–2.46) 0.005	1.55 (1.05–2.29) 0.03

Adjusted model adjusted for age, tumor size, nodes, ER status and current smoking status in addition to the other elements.

**Table 9 nutrients-15-02611-t009:** HRs associated with low and high serum metals and all-cause mortality (prostate cancer subjects).

	Total /Deaths	Age-Adjusted Only	Adjusted
		HR (95%CI) *p*-Value	HR (95%CI) *p*-Value
Selenium			
Low	51/24	1.91 (1.16–3.14) 0.01	1.57 (0.94–2.65) 0.09
Middle	231/54	1.0	1.0
High	65/11	0.84 (0.44–1.61) 0.59	1.02 (0.51–2.02) 0.96

Zinc			
Low	53/25	1.52 (0.93–2.48) 0.10	1.30 (0.77–2.19) 0.33
Middle	190/51	1.0	1.0
High	104/13	0.50 (0.27–0.93) 0.03	0.51 (0.27–0.96) 0.04

Copper			
Low	114/21	0.84 (0.49–1.42) 0.51	0.93 (0.54–1.59) 0.78
Middle	180/39	1.0	1.0
High	53/29	2.73 (1.67–4.44) <0.0001	2.71 (1.64–4.48) 0.0001

Adjusted model adjusted for age, PSA level, Gleason grade and current smoking status in addition to the other elements.

## Data Availability

Data supporting the reported results are available from the first author upon request from all interested researchers.

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
