# Peer review of "Serum Essential Elements and Survival after Cancer Diagnosis"

_nutrients, 2023, doi:10.3390/nu15112611_

Round 1

Reviewer 1 Report

Jan et al. report here a well-conducted study that found a high copper level was associated with an increase in all-cause mortality for all cancers combined. In contrast, a high zinc and selenium level was as sociated with reduced mortality. Among prostate cancer patients, a high serum copper level was strongly predictive of increased mortality; while among breast cancer patients, a low selenium level was associated with increased mortality. The topic is interesting. The experimental was well-designed. The study is compactly and well written, the figures and tables are well-presented. This reviewer, however, has a few suggestions that would improve the manuscript for readers.

1.     Lines 66-77, please add some new references for the three element research. Such as for selenium: 1) Selenium deficiency-induced multiple tissue damage with dysregulation of immune and redox homeostasis in broiler chicks under heat stress. SCIENCE CHINA-Life Sciences. 2023; 2) Both Selenium deficiency and excess impair male reproductive system via inducing oxidative stress-activated PI3K/AKT-mediated apoptosis and cell proliferation signaling in testis of mice. Free Radical Biology and Medicine. 2023, etc.

2.     Please add the approved information of the Experimentation Ethics Committee.

3.     Please add some references for the measurement of the elements in the methods section.

4.     Lines 134-136, please do not use only 2 rows as a paragraph. Please checking and revise the similar issues throughout the paper.

5.     Tables, please using the three lines for all the tables.

6.     Figures, please increase the resolution of all the figures.

7.     There are too many figures for this paper. Please combined them.

8.     Please checking the references and make sure all of them followed the journal style.

Author Response

We would like to thank the Editor for the opportunity to revise the manuscript. We have read the Reviewer’s 1 report and did our best to follow all valuable comments and suggestions to modify the manuscript accordingly. Our answers to the Reviewer’s comments are below:

Jan et al. report here a well-conducted study that found a high copper level was associated with an increase in all-cause mortality for all cancers combined. In contrast, a high zinc and selenium level was as sociated with reduced mortality. Among prostate cancer patients, a high serum copper level was strongly predictive of increased mortality; while among breast cancer patients, a low selenium level was associated with increased mortality. The topic is interesting. The experimental was well-designed. The study is compactly and well written, the figures and tables are well-presented. This reviewer, however, has a few suggestions that would improve the manuscript for readers.

  1. Lines 66-77, please add some new references for the three element research. Such as for selenium: 1) Selenium deficiency-induced multiple tissue damage with dysregulation of immune and redox homeostasis in broiler chicks under heat stress. SCIENCE CHINA-Life Sciences. 2023; 2) Both Selenium deficiency and excess impair male reproductive system via inducing oxidative stress-activated PI3K/AKT-mediated apoptosis and cell proliferation signaling in testis of mice. Free Radical Biology and Medicine. 2023, etc.

Ad. 1. References have been added.

  1. Please add the approved information of the Experimentation Ethics Committee.

Ad. 2. Scan of the Experimentation Ethics Committee 's consent added to the attachment (Please see the attachment).

  1. Please add some references for the measurement of the elements in the methods section.

Ad. 3. References have been added.

  1. Lines 134-136, please do not use only 2 rows as a paragraph. Please checking and revise the similar issues throughout the paper.

Ad. 4. Paragraphs have been changed.

  1. Tables, please using the three lines for all the tables.

Ad. 5. Tables have been changed.

  1. Figures, please increase the resolution of all the figures.

Ad. 6. The resolution of all figures has been increased.

  1. There are too many figures for this paper. Please combined them.

Ad. 7. Figures have been combined.

  1. Please checking the references and make sure all of them followed the journal style.

Ad. 8. References have been corrected.

Attention! Unexpectedly, we found technical error – our studies have been performed on serum and not on blood. Please accept appropriate changes.

Reviewer 2 Report

The topic presented in the manuscript is very interesting. The authors collected a very large number of samples in which the levels of zinc, copper and selenium in the blood of patients were determined, which I consider a great asset of this study. While the topic itself and its development are generally very good, a few elements caught my attention and I think they should be corrected/improved:

line 112 - in my opinion it would be better to write "Patients were asked" not, "study subjects".

Line 113 - you have written that whole blood were colllected into EDTA-filled tubes and after that it was frozen. Have you centrifudged probes? In my opinion, material and method section (measurment of elements and sample collection and storage) should be verified and rewritten. In usuall condition, blood samples are collected in a way that we collect 10ml from patients (into EDTA-filled tubes), centrifudge the solution, and collect supernatant (serum), that, serum alone is frozen and stored. Please, describe precisely and (consequentially, because now lines 112-114 and 116-121 contradict each other) how you collected blood samples for your experiment.

Measurement methodology - how many repeats were performed for each patients' serum? only 1 per 1 serum or more?

Figures - the quality of figures should be improved, especially in case of axis description and legend. I strongly advice to prepare figures in graphical programme or even in excel.

line 334 - "within" after this one,  no need to break the line

line 352 - remove quotation mark before "This"

Reference section - according to the MDPI reference list and citation style guide I strongly advice to, in positions 3,6,7,8,9,10,12,19,24,29,41, "cite the first ten authors, then add a semicolon and add ‘et al.’ at the end".

Author Response

We would like to thank the Editor for the opportunity to revise the manuscript. We have read the Reviewer’s 2 report and did our best to follow all valuable comments and suggestions to modify the manuscript accordingly. Our answers to the Reviewer’s comments are below:

The topic presented in the manuscript is very interesting. The authors collected a very large number of samples in which the levels of zinc, copper and selenium in the blood of patients were determined, which I consider a great asset of this study. While the topic itself and its development are generally very good, a few elements caught my attention and I think they should be corrected/improved:

line 112 - in my opinion it would be better to write "Patients were asked" not, "study subjects".

Ad. 1. It have been corrected.

Line 113 - you have written that whole blood were colllected into EDTA-filled tubes and after that it was frozen. Have you centrifudged probes? In my opinion, material and method section (measurment of elements and sample collection and storage) should be verified and rewritten. In usuall condition, blood samples are collected in a way that we collect 10ml from patients (into EDTA-filled tubes), centrifudge the solution, and collect supernatant (serum), that, serum alone is frozen and stored. Please, describe precisely and (consequentially, because now lines 112-114 and 116-121 contradict each other) how you collected blood samples for your experiment.

Ad. 2. Blood was donated both on the EDTA tubes and tube with clot activator. In this paper only the serum samples were measured.

As you recommended, this section was merged and re-written.

Measurement methodology - how many repeats were performed for each patients' serum? only 1 per 1 serum or more?

Ad. 3. All samples were measured once.

Figures - the quality of figures should be improved, especially in case of axis description and legend. I strongly advice to prepare figures in graphical programme or even in excel.

Ad. 4. Figures have been changed.

line 334 - "within" after this one,  no need to break the line

Ad. 5. It have been corrected.

line 352 - remove quotation mark before "This"

Ad. 6. It have been corrected.

Reference section - according to the MDPI reference list and citation style guide I strongly advice to, in positions 3,6,7,8,9,10,12,19,24,29,41, "cite the first ten authors, then add a semicolon and add ‘et al.’ at the end".

Ad. 7. References have been corrected.

Attention! Unexpectedly, we found technical error – our studies have been performed on serum and not on blood. Please accept appropriate changes.

Round 2

Reviewer 2 Report

The quality of the presented graphs could still be better, although in their current form the figures are definitely more legible than in the first version. Therefore, I accept them.
I don't have any more comments, I'm glad that the Authors noticed and corrected the errors mentioned earlier.

Author Response

We are very grateful for reviewing and accepting our manuscript.